# The Design, Kinematics and Torque Analysis of the Self-Bending Soft Contraction Actuator

**Alaa Al-Ibadi [1,2,*]** , **Samia Nefti-Meziani [1]** and **Steve Davis [1]**

[1]   Computing, Science and Engineering, Salford University, Salford M5 4WT, UK;
     S.Nefti-Meziani@salford.ac.uk (S.N.-M.); S.T.Davis@salford.ac.uk (S.D.)
[2]   Computer Engineering Department, Basrah University, Basrah 61004, Iraq
*   Correspondence: a.f.a.al-ibadi@edu.salford.ac.uk

**Abstract:** This article presents the development of a self-bending contraction actuator (SBCA) through the analysis of its structure, kinematics, and torque formulas, and then explores its applications. The proposed actuator has been fabricated by two methods to prove the efficiency of the human body inspiration, which represents the covering of human bones by soft tissues to protect the bone and give the soft texture. The SBCA provides bending behaviour along with a high force-to-weight ratio. As with the simple pneumatic muscle actuator (PMA), the SBCA is soft and easy to implement. Both the kinematics and the torque formula presented for the SBCA are scalable and can be used with different actuator sizes. The bending actuator has been tested under an air pressure of up to 500 kPa, and the behaviour of its bending angle, parameters, dimensions, and the bending torques have been illustrated. On the other hand, the experiments showed the efficient performances of the actuator and validate the proposed kinematics. Therefore, the actuator can be used in many different applications, such as soft grippers and continuum arms.

**Keywords:** soft robotics; pneumatic muscle actuators (PMA); self-bending contraction actuator (SBCA); kinematics; torque formula

---

## 1. Introduction

For various reasons, during the last two decades, researchers have focused their attention on developing new actuators for industrial, medical and academic use. Influencing factors for this include the requirement to decrease the cost of the systems and increase the range of applications. The soft actuator is one of the recent developments in this area, and the pneumatic muscle actuator (PMA) represents a valuable example. The advantages of the PMA encourage researchers to use and enhance the performance of the air actuator [1]. These advantages include its high force-to-weight ratio, low cost, easy implementation and deformation, lightweight and the safety of human beings working close to them [2–4]. The PMA is constructed from an inner tube surrounded by a braided mesh closed on both sides and with a small air inlet. The braided angle ($\theta$) defines the operation of the actuator and its critical value is 54.7°. Contraction behaviour occurs when $\theta$ is less than the critical value, while the higher braided angle creates an extension behaviour for the air muscle [5–7].

The tensile force for the contraction actuator has recently been presented and modified by numerous researchers [2,4,8–10]. Furthermore, the extension force has also been presented as a modified form of the tensile force by [11].

The bending pneumatic air muscles have been developed by numerous researchers. Among them is that presented by [12], using two chambers instead of one inner tube to create a bending actuator. The authors in [13] proposed a PneuNet actuator by utilising various thicknesses for the inner tube. The implementation of the PneuNet is easy, however, it lacks in the ratio of the elasticity.

To overcome this restriction, a polymer fibre is used in the PneuFlex actuator to support the rubber substrate, as proposed by [14]. In comparison with the silicone, the polyethylene terephthalate (PET) material is three to four times less elastic. Notably, reference [15] utilised the strain limiting layer on one side of the extension actuator to prevent elongation from this side and to make the other side free to extend. This method is used by [2], but adapted by using a high tension thread to partially fix the extensor PMA length. Furthermore, reference [16] utilised the impact of the braided angle by using two different braided angles of the braided mesh for the contractor muscle actuator to develop the bending performance. Moreover, reference [17] used two integrated jamming actuators in parallel to create bending performance. The jamming technique is used by [18] to control the bending stiffness.

Notably, references [19,20] used tendons to establish the bending behaviour for the soft gripper. The PMA can be used in various engineering areas, including humanoid robots, wearable robots for medical applications, industrial and airspace applications, and mobile robots [21]. On the other hand, bending muscle can be implemented by 3D printing; this technology provides fast prototyping, flexible design, and an easy way to implement the actuators and sensors that have the complex structure [22]. Furthermore, 3D printing offers an efficient way to build an actuator by using different materials and layouts at the same time [23]. A fused deposition modelling (FDM) technology has been used by [24] to develop a 3D printed pneumatic muscle. Inserting soft sensors during the fabrication of the pneumatic actuators by 3D technology is called the 4D printing method, and it provides a valuable technique to manufacture such types of actuators [25].

In this article, the construction of the self-bending contraction actuator self-bending contraction actuator (SBCA) is presented in detail, together with its performance. The kinematics and the bending torque formulas are proposed and validated for SBCAs with different dimensions.

The main contributions of this article are the proposed kinematics for the SBCA and the bending torque formula. The paper is organised as follows: Section 2 details the construction of the SBCA and the materials used, together with the components of the experiment to show its performance. The kinematics for the proposed actuator are presented in Section 3, the bending torque is proposed in Section 4, while the validations and the possible applications for the SBCA are explained in Sections 5 and 6, respectively.

## 2. The Self-Bending Contraction Actuator (SBCA)

The behaviour of the soft muscle is either contraction or extension, and it is defined by the construction of the actuator and its dimensions. The elementary construction of the air muscle is shown in Figure 1. The size for both the inner rubber tube and the braided cover mesh defines the initial length $L_0$ and initial diameter $D_0$ for the PMA. The inactive unpressurised value of the braided angle ($\theta$) controls whether the actuator will act as a contraction or extension muscle. If $\theta$ is less than 54.7°, the PMA is a contraction actuator, and it behaves as an extension actuator if the resting braided angle is more than 54.7° [9,26,27].

Both the contraction ratio and the extension ratio depend on the structure of the PMA, which defines the braided angle, the stiffness and the dimensions of the inner tube, including its thickness and diameter, as well as the maximum diameter of the braided sleeve [4]. Equation (1) defines the contraction ratio and (2) describes the extension ratio, respectively.

$$\varepsilon = \frac{L_0 - L}{L_0} \tag{1}$$

$$\acute{\varepsilon} = \frac{L - L_0}{L} \tag{2}$$

where $\varepsilon$ is the contraction ratio for the contractor PMA and $\acute{\varepsilon}$ represents the extension ratio for the extensor actuator.

According to the explanation of [28] for the constant-volume principle, the dimensional change in this part of the actuator leads to the dimensional adjustment on another side, and that creates the bending behaviour.

The PMA is traditionally used to behave as a contraction or extension actuator. Nevertheless, this paper introduces a bending behaviour for the contraction type pneumatic muscle actuators.

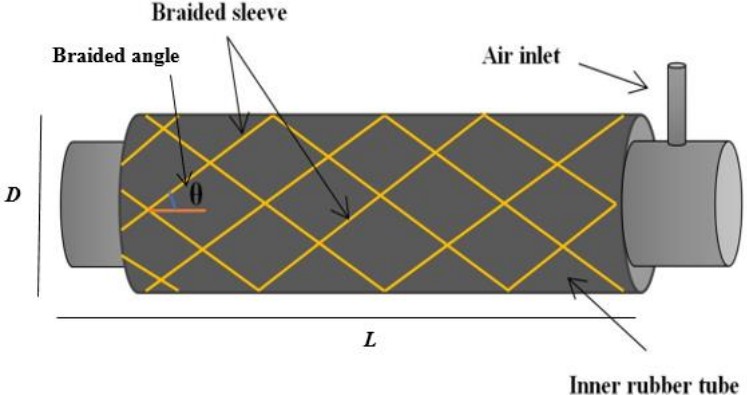

**Figure 1.** The structure of the pneumatic muscle actuator.

*Fabrication of the SBCA*

Practically, the pneumatic muscle actuator is fabricated by using an inner rubber tube, a mesh sleeve and two solid ends with an air inlet [4]. In this section, a thin (2 mm) flexible but incompressible reinforcing rod is used to partially fix the length of the contraction actuator.

Firstly, the rod is 3D printed with numerous small holes along its length, and it is sewn on the outer covering of the actuator, as shown in Figure 2. Applying air pressure forces the actuator to contract from all the free sides, while the rod prevents the contraction along its length. This method gives the bending behaviour of a single contraction actuator.

As shown in Figure 2, the actuator, applying a high force on the rod, leads to breaking, after several bending iterations. Moreover, the external rigid rod decreases the soft appearance of the PMA. On the other hand, we were inspired to copy the idea of our body for biological concepts. The rigid parts in the human body (bones) are covered by soft tissue (skin), and that makes the external appearance soft. Therefore, the reinforcement rod is inserted between the inner rubber tube and the braided sleeve. This method achieves the biological concept and increases the durability of the rod, because both the rubber tube and the covered shell apply force from both sides. Figure 3 illustrates the SBCA with the new construction approach.

With the 30 cm actuator, an experiment was conducted to study the bending angle at different values of the attached load, as follows:

1. Fix the SBCA from the air inlet side vertically.
2. Connect a 9-axis motion tracking (BNO055) sensor to the free end, to measure the bending angle.
3. Attach load support to adjust the load value.
4. Apply air pressure via (3/3 solenoid valve).
5. Measure the air pressure by a pressure sensor.
6. Use Arduino Mega 2560 to control the experiment process.

The BNO055 sensor is attached to the free end of the SBCA, while the other end is fixed by a screw. Euler angles formula is used to calculate the orientation (bending) angle and it is calibrated to be zero and no bending (zero applied pressure).

The experimental components are shown in Figure 4, and Table 1 lists the maximum bending angle at different loads.

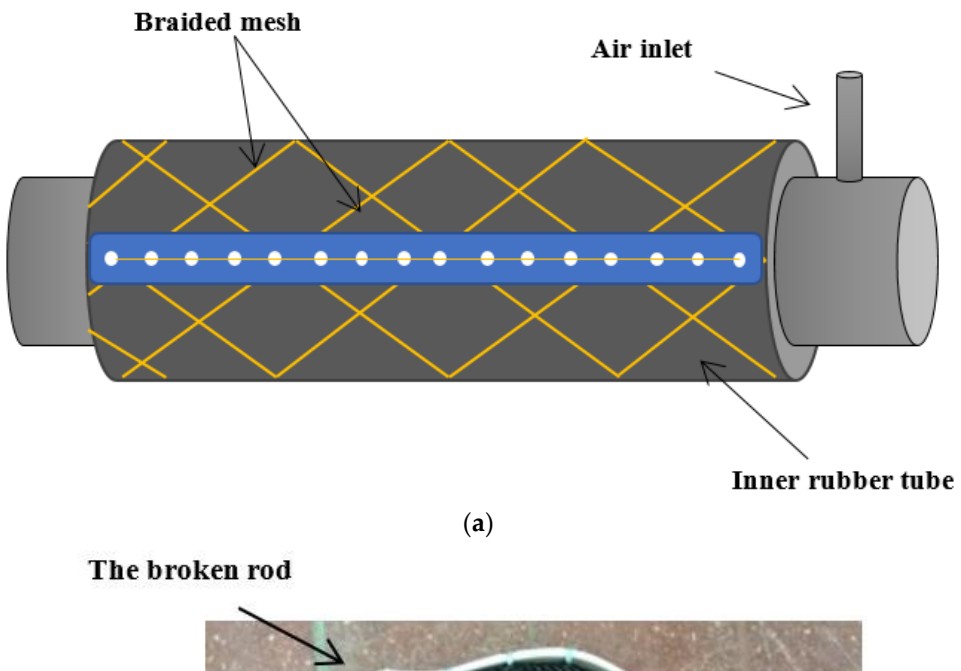

(a)

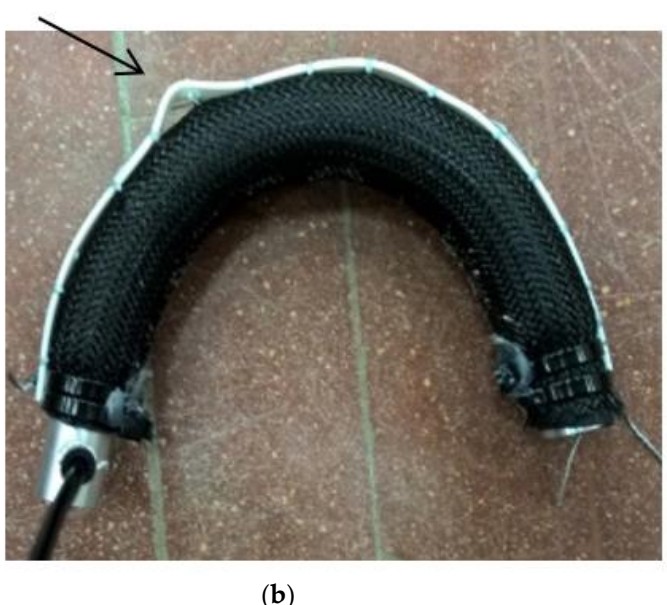

(b)

**Figure 2.** The structure of the self-bending contraction actuator version I. (**a**) The layout of the reinforcement rod. (**b**) The bending actuator after several operations.

**Table 1.** The maximum bending angle at different loads for 30 cm SBCA.

| Load (kg) | Bending Angle (Degree) |
|:---------:|:----------------------:|
| 0.0 | 213.1 |
| 0.5 | 136.2 |
| 1.0 | 73.0 |
| 1.5 | 49.3 |
| 2.0 | 34.1 |

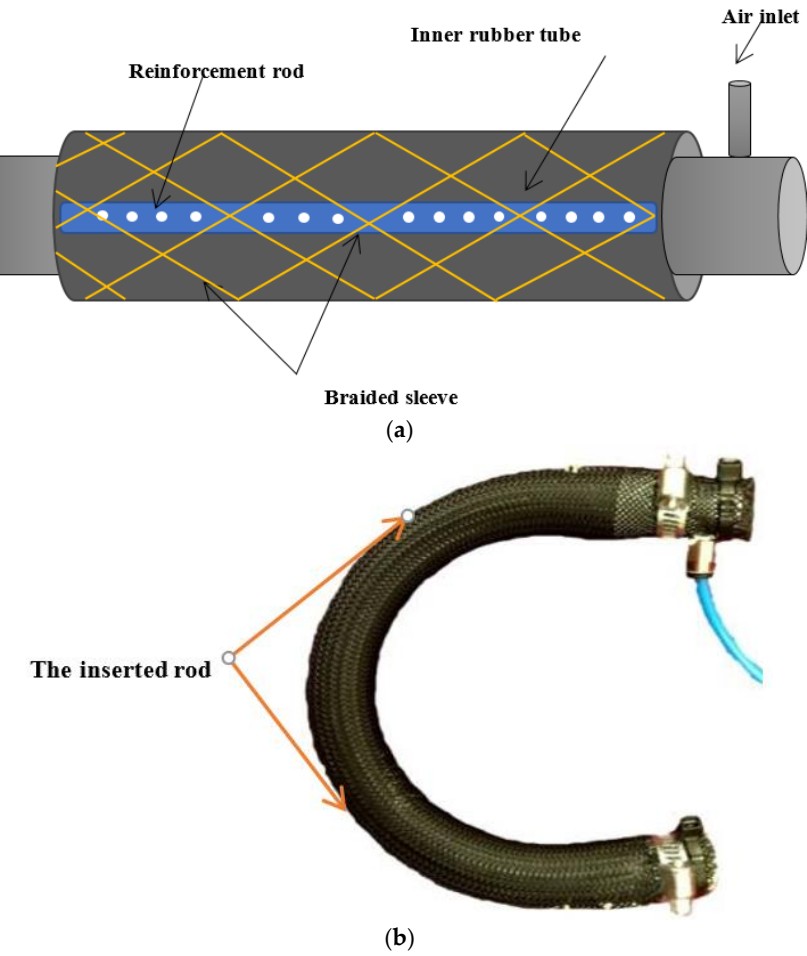

**Figure 3.** The novel structure of the self-bending contraction actuator (SBCA) version II. (**a**) The structure of the SBCA showing the inserted rod. (**b**) The 30 cm SBCA at 300 kPa.

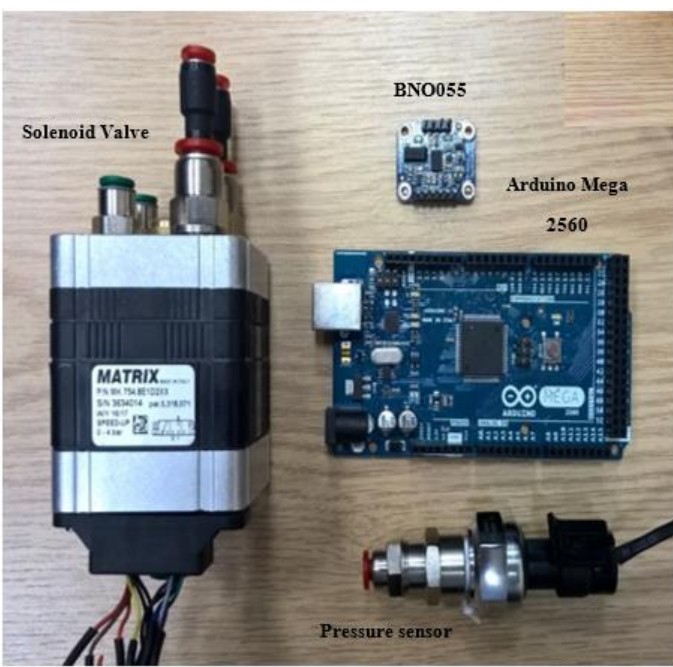

**Figure 4.** The experiment components.

Figure 5 shows the bending angle as a function of air pressure at different load values. Furthermore, the dimensions of the used material in this bending actuator are listed in Table 2.

**Table 2.** The dimensions of the bending pneumatic muscle actuator (PMA).

| $L_0$ (m) | Rubber Thickness (m) | Braided Thickness (m) | Inner Diameter (m) |
|---|---|---|---|
| 0.3 | $1.1 \times 10^{-3}$ | $0.5 \times 10^{-3}$ | $12 \times 10^{-3}$ |
| **Rubber stiffness (N/m)** | **Rod length (m)** | **Rod thickness (m)** | **Rod width (m)** |
| 363.33 | 0.3 | 0.002 | 0.016 |

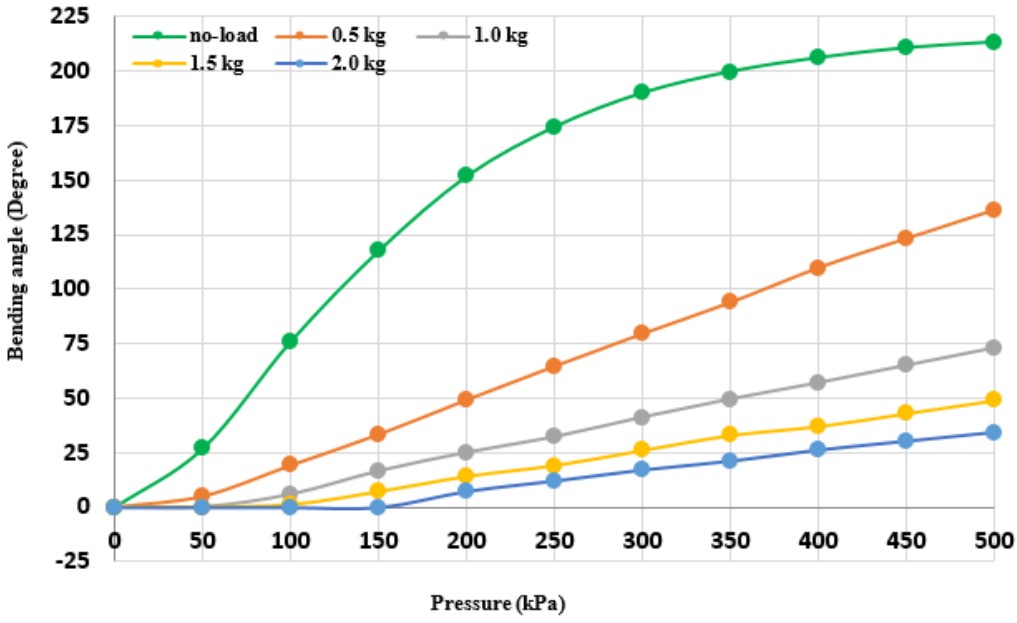

**Figure 5.** The bending angle at different attached loads for the 30 cm SBCA.

## 3. The Kinematics of the SBCA

In general, connecting the two ends of a line with length $L$ results in a circle shape (Figure 6). The dimension for this circle can be calculated as:

$$circumference = L = \pi D = 2\pi r \tag{3}$$

where: $D$ is the diameter and $r$ is the radius of the circle.
Or:

$$D = \frac{L}{\pi} \tag{4}$$

Or:

$$r = \frac{L}{2\pi} \tag{5}$$

The angle of the circle is fixed at 360°.

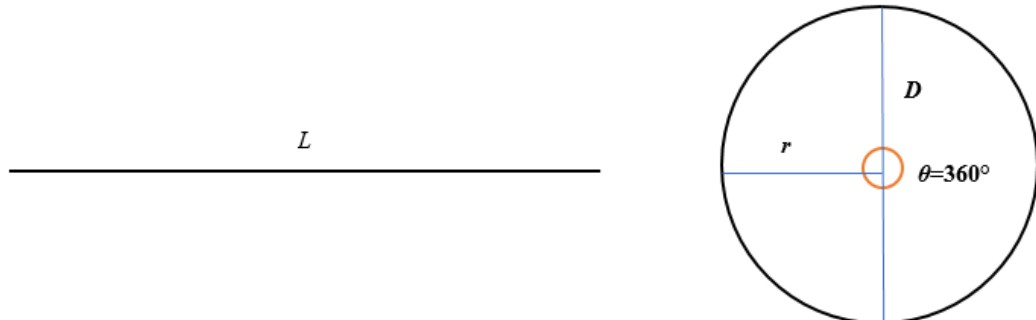

**Figure 6.** The line to circle conversion.

Since the maximum bending angle of the SCBA is not constant and depends on the resting length $L_0$ of the actuator, the bending angle can be measured using the *Arc* length formula as follows:

$$\beta° = \frac{360° \, Arc}{2\pi r} \tag{6}$$

Figure 7 illustrates the geometrical analysis of the SBCA's bending behaviour.

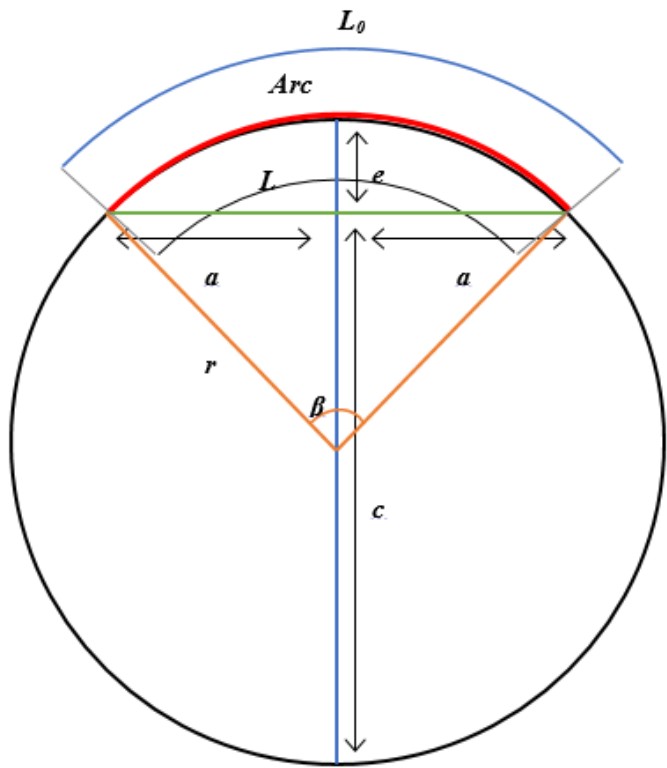

**Figure 7.** The geometrical analysis of the SBCA.

Since the rear length $L_0$ is constant due to the reinforcement rod, the only change will be in the length $L$, because of the contraction performance. The bending angle $\beta$ of the SBCA will be calculated according to the middle *Arc* (see Figure 7) as follows:

$$Arc = \frac{L_0 + L}{2} \tag{7}$$

From (1):

$$L = L_0(1 - \varepsilon) \tag{8}$$

Substituting (8) in (7):

$$Arc = \frac{L_0(2 - \varepsilon)}{2} \tag{9}$$

And:

$$\beta° = \frac{360° \, L_0(2 - \varepsilon)}{4\pi r} \tag{10}$$

From (10), the bending angle $\beta$ depends on the initial length of the actuator $L_0$, the contraction ratio, and the radius. From the previews, the maximum contraction ratio is about 30%.

Hence, from (10) and Figure 7, the maximum bending angle $\beta_{max}$ occurs at the maximum contraction ratio $\varepsilon_{max}$ and the minimum radius $r_{min}$ for a constant initial length.

Then:

$$\beta_{max}° = \frac{360° \, L_0(2 - \varepsilon_{max})}{4\pi r_{min}} \tag{11}$$

Or:

$$\beta_{max}° = \frac{48.7 L_0}{r_{min}} \tag{12}$$

Again, from Figure 7 and using the intersecting chords theorem [29]:

$$a.a = e.c \tag{13}$$

Alternatively:

$$c = \frac{a^2}{e} \tag{14}$$

The diameter of the circle is:

$$Diameter = e + c \tag{15}$$

Or:

$$Diameter = e + \frac{a^2}{e} \tag{16}$$

Then, the radius is:

$$r = \frac{a^2 + e^2}{2e} \tag{17}$$

Since $a$ is half of the $Arc$'s width $W$, and $e$ is the $Arc$'s height $H$, then:

$$r = \frac{W^2 + 4H^2}{8H} \tag{18}$$

From (18), the radius of the $Arc$ can be found from its height and width.

Special case 1:

As $\beta$ is equal to 360°, the SBCA shapes like a circle. Therefore, the width of the $Arc$ is zero, the radius is constant, and it is equal to $H/2$.

Special case 2:

If the $W$ is zero, the radius is constant and equals $H/2$, and $\beta$ is equal or more than 360°, the value of the bending angle only depends on the initial length of the SBCA.

## 4. The Bending Torque of the SBCA

The volume of the cylinder is:

$$V = \frac{\pi D^2 L}{4} \tag{19}$$

From Figure 8, the volume of the SBCA can be defined as:

$$V = \frac{\pi D^2 Arc}{4} \tag{20}$$

And:

$$Arc = b\ cos\theta \tag{21}$$

$$D = \frac{b\ sin\theta}{n\pi} \tag{22}$$

Then:

$$V = \frac{b^3\ sin^2\theta\ cos\theta}{4n^2\pi} \tag{23}$$

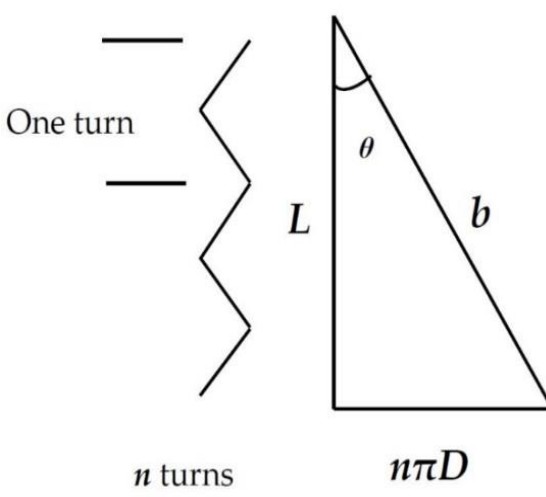

**Figure 8.** The parameters of the PMA.

From the virtual work theorem, the total force ($F_t$) can be calculated as:

$$F_t = P\ \frac{dV}{dArc} \tag{24}$$

Or:

$$F_t = P\ \frac{b^2}{4n^2\pi}\left(sin^2\theta - 2cos\theta\right) \tag{25}$$

The increment of the actuator diameter causes a radial force on the inserted rod; this force can be calculated by using the virtual work theorem as follows:

$$F_{rt} = P\frac{dV}{dD} \tag{26}$$

Or:

$$F_{rt} = P\frac{b^2}{4ncos\theta}\left(2sin\theta\ cos\theta - sin^3\theta\right) \tag{27}$$

where: $F_{rt}$ is the total resistance force. The circumference of the actuator is:

$$C = n\pi D \tag{28}$$

Since the resistance force affects only the rod, the resistance force on the rod is:

$$F_{rr} = \frac{F_{rt}}{\left(\frac{C}{W_r}\right)} \tag{29}$$

where: $W_r$ is the width of the rod, the net torque of the SBCA can be calculated as:

$$T = (F_t - F_{rr}) * Arc \tag{30}$$

From (30), for similar materials, the torque of the SBCA depends on its length.

To validate the proposed kinematics, three different bending actuators are designed and built with the initial lengths of 20 cm, 30 cm, and 50 cm, respectively. Then, the air pressure of 500 kPa is applied, to record the bending angle for each actuator by the BNO055 sensor. For measuring the bending force, a load is attached to the end of the SBCA and the load is increased until the bending angle becomes zero at maximum air pressure (500 kPa). The calculated torque can be found by using (24) and (26)–(30). Figure 9 shows the three actuators at 500 kPa and the 30 cm SBCA at maximum load. Table 3 gives the dimensions of the *Arc* and Table 4 lists the bending angle and the bending torque for the three actuators.

**Table 3.** The dimension of the *Arc*.

| $L_0$ (cm) | $W$ (cm) | $H$ (cm) | $R$ (cm) | *Arc* (cm) |
|---|---|---|---|---|
| 20 | 14 | 6 | 7.08 | 17 |
| 30 | 11.5 | 10.2 | 6.72 | 25.5 |
| 50 | 0 | 10.5 | 5.25 | 42.5 |

**Table 4.** The measured and calculated bending angle and bending torque of the three various SBCAs.

| $L_0$ (cm) | Measured Bending Angle (Degree) | Calculated Bending Angle (Degree) | Measured Bending Torque (Nm) | Calculated Bending Torque (Nm) |
|---|---|---|---|---|
| 20 | 135 | 137.5 | 45.126 | 44.27 |
| 30 | 215 | 217.38 | 68.67 | 66.48 |
| 50 | 462 | 463.8 | 107.91 | 110.09 |

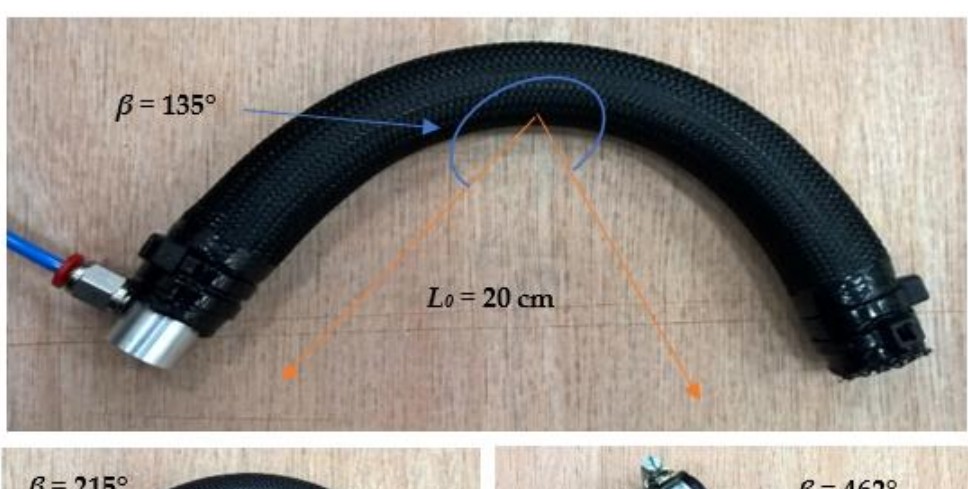

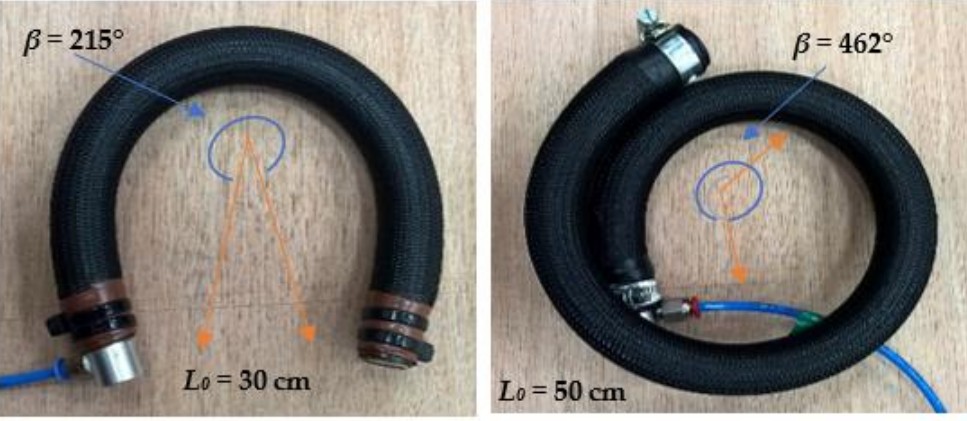

(a)

**Figure 9.** *Cont.*

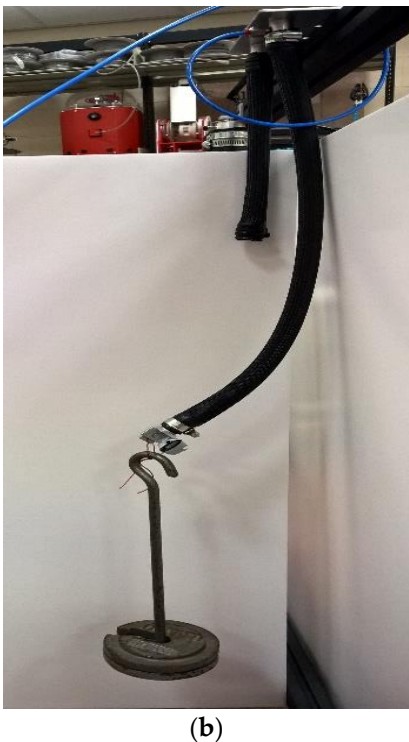

(**b**)

**Figure 9.** Three SBCAs at 500 kPa of various lengths (**a**) at no-load. (**b**) The 30 cm SBCA at 300 kPa and 1.0 kg.

Figure 10 illustrates the measured and the theoretical (10) bending angle of the 30 cm actuator as a function of air pressure, and Table 5 shows the width, the height, and the radius of the *Arc* at air pressures from 0 kPa to 500 kPa.

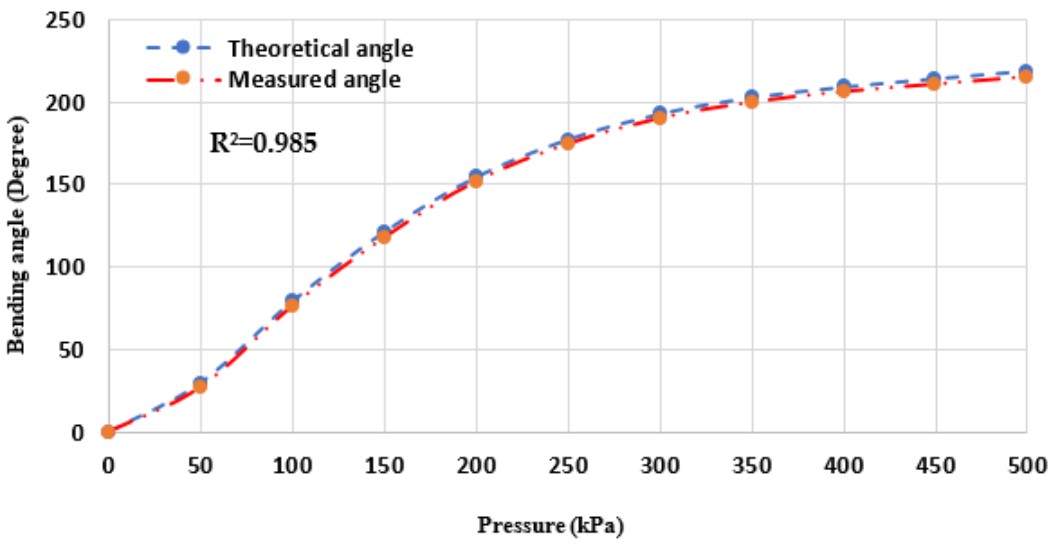

**Figure 10.** The bending angle of the 30 cm SBCA.

**Table 5.** The dimensions of the *Arc* of the SBCA at different air pressures.

| Pressure (kPa) | Width (cm) | Height (cm) | Radius (cm) |
|:---:|:---:|:---:|:---:|
| 0 | 30.0 | 0.0 | - |
| 50 | 29.5 | 1.9 | 58.20 |
| 100 | 26.9 | 4.9 | 20.91 |
| 150 | 24.1 | 7.9 | 13.14 |
| 200 | 20.1 | 9.4 | 10.07 |
| 250 | 17.2 | 9.8 | 8.67 |
| 300 | 15.3 | 9.9 | 7.90 |
| 350 | 14.1 | 9.95 | 7.47 |
| 400 | 13.3 | 10.0 | 7.21 |
| 450 | 12.6 | 10.1 | 7.04 |
| 500 | 11.5 | 10.2 | 6.72 |

Table 5 shows that the radius at zero pressure is undefined. The SBCA at no applied pressure behaves like a line (zero bendings) and the radius for a line is infinity (see (6) and (18)).

Equations (19)–(30) show the calculation requirements to find the bending torque. To validate these equations, a 30 cm SBCA is used for the parameters listed in Table 6.

**Table 6.** The Parameters of the 30 cm SBCA.

| $L_0$ (cm) | $b$ (cm) | $n$ | $W_r$ (cm) |
|:---:|:---:|:---:|:---:|
| 30 | 34.64 | 3.15 | 1.2 |

Figure 11a–c show the behaviour of the *Arc*, the threaded angle, and the bending torque at various applied pressures.

Figure 11a shows that the *Arc* is behaving like the length of the contraction PMA, but at a lower contraction ratio, due to the preventing length change at the rod side. Figure 11b gives the normal performance of the braided angle for the contraction PMA [9,30]. Figure 11c shows that the variations of the bending torque after 450 kPa are small, because of the small change in the dimensions of the SBCA.

In comparison with the bending actuator of [12], the SBCA works under more air pressure, 500 kPa against 150 kPa, and provides a higher bending angle. Similar positives have been noticed over the bending actuator having two chambers described by [16]. On the other hand, the bending behaviour of [20] is too low, and it depends on the contraction ratio of the contractor PMA, in addition to the complexity of the system.

Apart from the bending angle, the SBCA provides high bending torque in comparison with its weight (40 g, 55 g, and 75 g for 20 cm, 30 cm, and 50 cm actuators respectively). Furthermore, the experiments show that the bending angle of the proposed actuator increases when the initial length of the actuator increases. The bending torque also depends on the initial length of the actuator. On the other hand, the width of the inserted rod is a major factor for the torque of the SBCA, because the decrease in width causes less resistance force and that leads to increasing the net torque of the actuator.

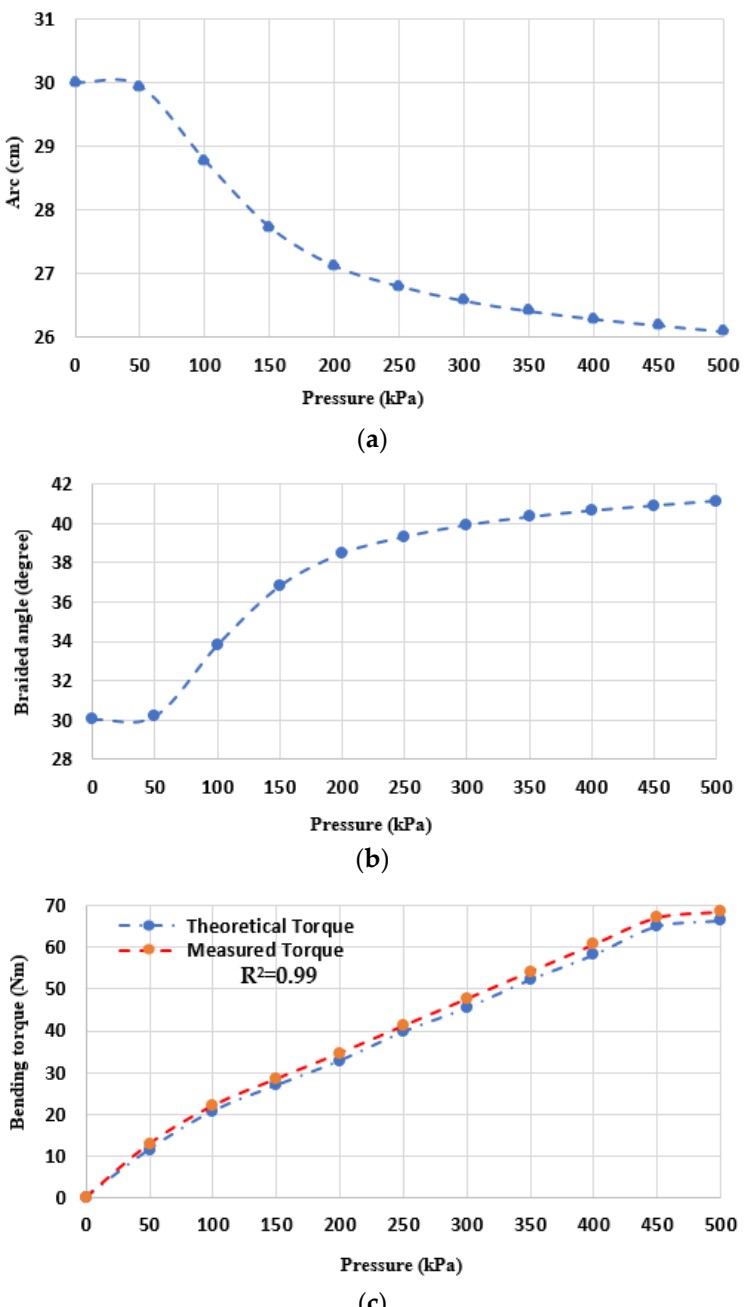

**Figure 11.** The performance of the 30 cm SBCA. (**a**) The behaviour of the *Arc* length. (**b**) The variation of the braided angle. (**c**) The bending torque as a function of air pressure.

## 5. Applications of the SCBA

The bending behaviour of the SBCA makes it suitable for different applications, for example as a bending finger. A soft gripper can be made from several fingers, as shown in Figure 12. Each gripper is formed from a series of SBCA arranged around a fixed rigid palm. The SBCA only generates a force in a single direction an elastomeric ribbon is attached to the rear of each finger to extend it when pressure is released. The grippers have been demonstrated grasping a range of everyday items and the grippers with more fingers were found to be able to grasp objects with much more complex geometry. Unlike a traditional rigid gripper, the soft SBCA finger naturally deforms around the object to be grasped, maximising the stability of the grasp without the need for complex grasp planning or control.

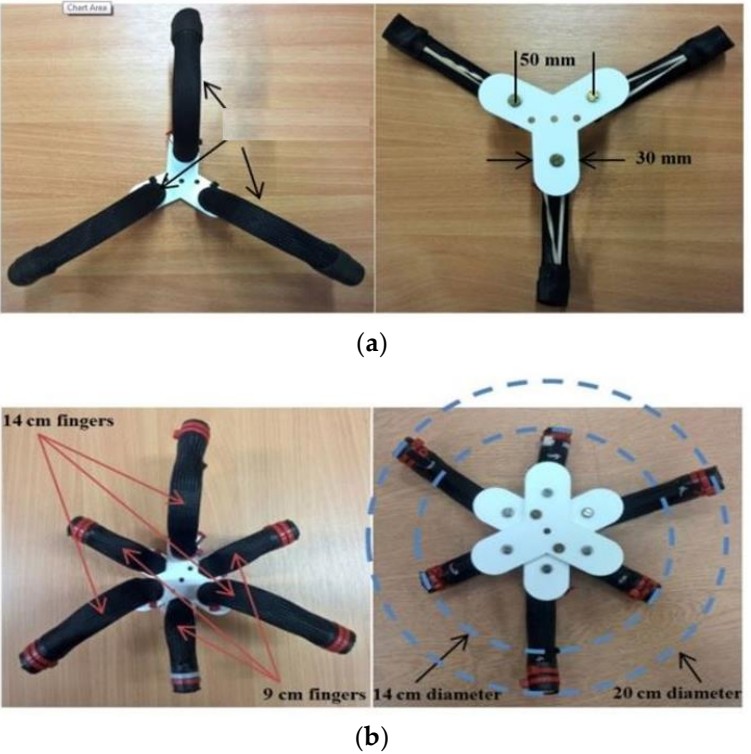

**Figure 12.** Three and six finger soft grippers using the SBCA [3]. (**a**) The three fingers soft gripper. (**b**) The two grasping-point rows, six fingers soft gripper.

The SBCA can also be used as a continuum robot arm. Since both the bending angle and the bending payload depend on the dimensions of the SBCA, various dimensions of continuum arms can be made for different applications. Figure 13 shows two continuum arms made from SBCAs; the first one is made from a single SBCA and allows for very simple manipulation tasks, for example where objects need to be moved between two fixed locations. It can be seen in Figure 13a that both the ball and measuring tape can be grasped by the two-finger soft gripper and repeatably move to the same location. The second soft arm allows for much more complex positioning and movement of the object being grasped, as it is made from two bending actuators arranged in series but having different orientations. This configuration allows each section of the arm to bend in different directions, and by combining the bending of the two-section, the range of possible end positions that the arm can achieve can be greatly increased. As can be seen in Figure 13b, this system allows an object to be grasped and then moved to a range of different locations.

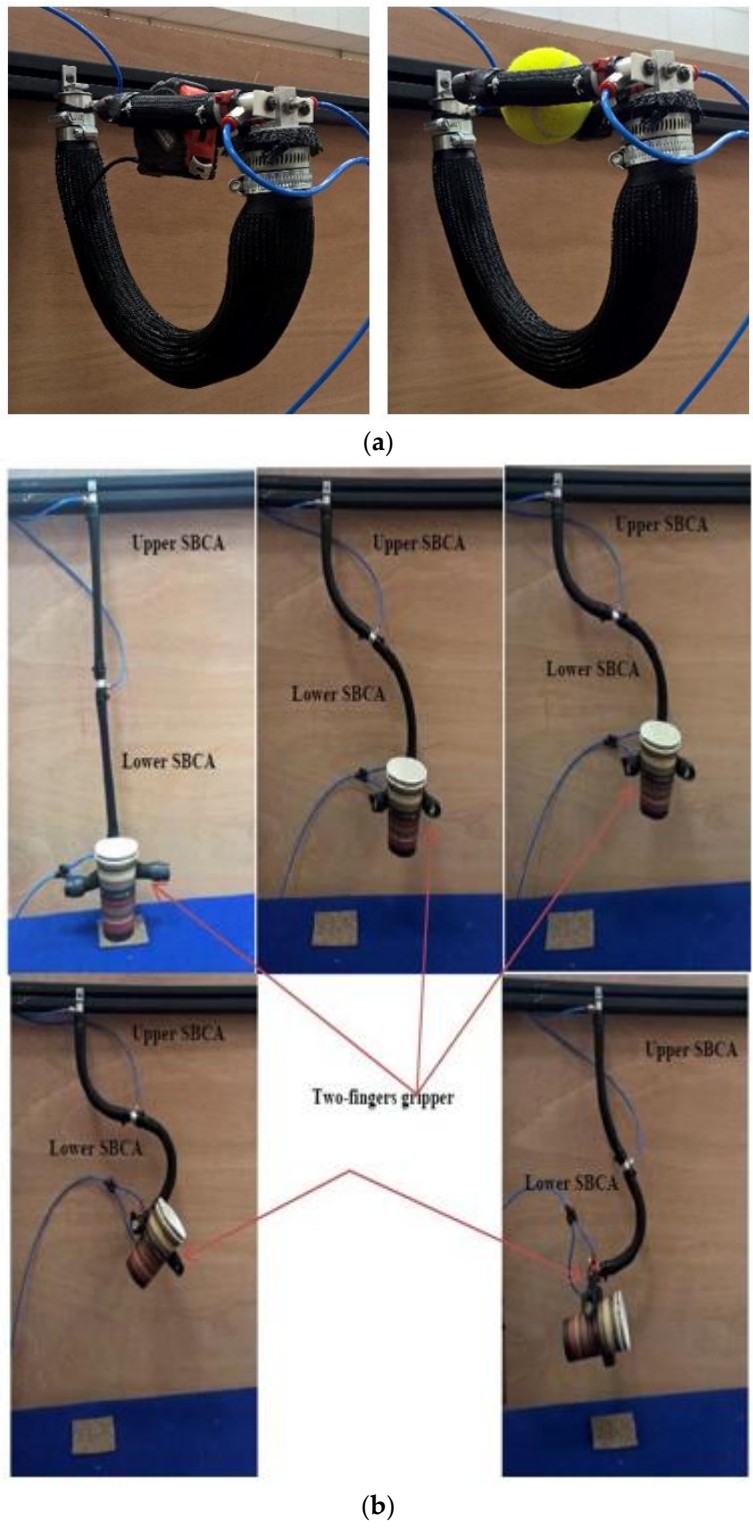

**Figure 13.** Continuum arms based on SBCA and a two-finger soft gripper. (**a**) A single SBCA. (**b**) Two SBCAs in opposite bending directions.

## 6. Conclusions

In this article, a new type of soft pneumatic actuator is proposed. The main feature of the proposed self-bending contraction actuator (SBCA) is its efficient bending behaviour. The inserted flexible rod between the braided mesh and the inner tube establishes the bending behaviour and provides a high payload in comparison with the actuator's weight. The SBCA is better than other bending actuators,

due to the ability to resize and replace the inserted rod, which leads to changes in the performance of the actuator. This is in contrast to the materials used by the other bending actuators being fixed, either by using two different braided angles of the outer coverage or using a high thickness of the tube wall.

This paper presented two techniques for providing a bending performance; the first version of the SBCA failed after several trials because of damage to the rod when it was fixed on the outer side of the actuator. In addition to this damage, this technique also reduces the softness of the system as the rigid material located on its exterior. In order to solve this issue, the flexible rod has been inserted between the braided mesh and the inner tube. By using this method, both the inner rubber tube and the braided cover apply force on the rod in both directions, which prevents damage.

The kinematics and the torque generated have also been analysed and the bending angle, and the torque formulas have been validated for three different actuator lengths; 20 cm, 30 cm, and 50 cm. The SBCA can be used in many applications, such as soft grippers, soft robot arms, and applications where bending motion is required.

**Author Contributions:** A.A.-I. designed and performed the experiments; A.A.-I. and S.D. analysed the data; A.A.-I. wrote the paper; and S.N.-M. edited the paper. All authors have read and agreed to the published version of the manuscript.

**Funding:** This research received no external funding.

**Acknowledgments:** The authors would like to thank the Ministry of Higher Education, Iraq, and the University of Basrah's Computer-Engineering Department for providing scholarship support to the first author of this paper.

**Conflicts of Interest:** The authors declare no conflict of interest.

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
