# Peer review of "The Design, Kinematics and Torque Analysis of the Self-Bending Soft Contraction Actuator"

_actuators, doi:10.3390/act9020033_

Round 1

Reviewer 1 Report

Some comments

As Fig 5 is an experimental results,  all data points should be shown and lines should only be curve fitted and not jointed.

As the authors have developed the mathematical model of the actuator, the results should be placed together with experimental data to justify that the kinematics is correct.  Further, mentioned it has developed the torque formulas, and I would expect that simulation results to be shown in the manuscript.

The results tabulated in Table 4 are too limited and does not prove that the formulation agrees with the empirical results.

What is the force that can be generated by the actuators.  Photos evidence as in Fig 9 to 11 do not prove anything with regards to the performance and characteristics of the proposed actuators but merely to show that the authors have conducted some experiments.

In summary, the presented results lack sufficient justification. I recommend that the authors revise the manuscripts with significant amount of results in order to justify their work. In this present form, it is not convincing at all,

Reviewer 2 Report

Dear editor,

The manuscript presents a soft pneumatic actuator demonstrating self-bending contraction. The kinematics and the torque generated have were analysed and the bending angle and the torque formulas were validated for three different actuator lengths. The article has shown some acceptable investigations, however, it needs more explanations and details to be ready for the publication.

  1. The abstract should be extended including more specific details of work done in the manuscript.
  2. A new section should be included describing the details of manufacturing the actuator, including the material and fabrication steps.
  3. More explanations of how and type of the rod are sewn on the outer covering of the actuator shown in Fig. 2 should be provided.
  4. How the motion tracking (BNO055) sensor was calibrated for measuring the bending of the actuator and where and how it was installed. These should be included in the manuscript.
  5. The literature review should be elaborated more on the application of soft robots and actuators. The following papers could be a clue to refer:

Evolution of 3D printed soft actuators. Sensors and Actuators A: Physical, 2016, 250: 258-272.

3D printing for soft robotics–a review. Science and technology of advanced materials 19, no. 1 (2018): 243-262.

Closed-loop 4D-printed soft robots

High-force soft printable pneumatics for soft robotic applications. Soft Robotics 3, no. 3 (2016): 144-158.

  1. Several typo errors should be revised throughout the manuscript, such as:

Page 2 Line 73 “Where: ε is the contraction” there is no need to ‘:’ after where.

Round 2

Reviewer 1 Report

The authors have not addressed the comments provided.  The work is incomplete without addressing these concerns.  Hence, unless it is addressed, I do not see much contribution of this work and have to reject it.

Reviewer 2 Report

The authors have addressed my earlier main concerns. However, still, the abstract should be improved enumerating the details of what has done in the paper rather than general speaking, before sending for the production.

Regards.

Author Response

Thank you for your comments. I have updated the abstract to include more details about the experiments and results.

Round 3

Reviewer 1 Report

Thank you for the revision.  However, the torque as presented in Fig 11 needs to compare the theoretical versus experimental results to validate that the numerical model relating to the torque is valid. 

Further,as mentioned in my previous review, that experimental results MUST show all the datapoints. This allows the reader to know exactly the at what parameter values the data is collected.  A smooth line graph is good ONLY for theoretical NUMERICAL work.

Author Response

* Figure 11.a-c has been updated as suggested.
